# *piggy*Bac Transposon-Based Immortalization of Human Deciduous Tooth Dental Pulp Cells with Multipotency and Non-Tumorigenic Potential

**DOI:** 10.3390/ijms20194904

**Published:** 2019-10-03

**Authors:** Emi Inada, Issei Saitoh, Naoko Kubota, Yoko Iwase, Yuki Kiyokawa, Shinji Shibasaki, Hirofumi Noguchi, Youichi Yamasaki, Masahiro Sato

**Affiliations:** 1Department of Pediatric Dentistry, Graduate School of Medical and Dental Sciences, Kagoshima University, Kagoshima 890-8544, Japan; inada@dent.kagoshima-u.ac.jp (E.I.); kubonao@dent.kagoshima-u.ac.jp (N.K.); yamasaki@dent.kagoshima-u.ac.jp (Y.Y.); 2Division of Pediatric Dentistry, Graduate School of Medical and Dental Science, Niigata University, Niigata 951-8514, Japan; isaito@dent.niigata-u.ac.jp (I.S.); iwase@dent.niigata-u.ac.jp (Y.I.); ykiyokawa@dent.niigata-u.ac.jp (Y.K.); 3Faculty of Dentistry, Niigata University, Niigata 951-8514, Japan; Cherryblossoms0313@gmail.com; 4Department of Regenerative Medicine, Graduate School of Medicine, University of the Ryukyus, Okinawa 903-0215, Japan; noguchih@med.u-ryukyu.ac.jp; 5Section of Gene Expression Regulation, Frontier Science Research Center, Kagoshima University, Kagoshima 890-8544, Japan

**Keywords:** immortalization, telomerase reverse transcriptase, E7, human deciduous tooth-derived dental pulp cell, *piggy*Bac, multi-differentiation potential, tumorigenicity, stemness factor, alkaline phosphatase

## Abstract

We aimed to immortalize primarily isolated human deciduous tooth-derived dental pulp cells (HDDPCs) by transfection with *piggy*Bac (PB)-based transposon vectors carrying *E7* from human papilloma virus 16 or complementary DNA (cDNA) encoding human telomerase reverse transcriptase (*hTERT*). HDDPCs were co-transfected with pTrans (conferring PB transposase expression) + pT-pac (conferring puromycin acetyltransferase expression) + pT-tdTomato (conferring *tdTomato* cDNA expression) and pT-E7 (conferring *E7* expression) or pTrans + pT-pac + pT-EGFP (conferring enhanced green fluorescent protein cDNA expression) + pT-hTERT (conferring *hTERT* expression). After six days, these cells were selected in medium containing 5 μg/mL puromycin for one day, and then cultured in normal medium allowing cell survival. All resultant colonies were harvested and propagated as a pool. Stemness and tumorigenic properties of the established cell lines (“MT_E7” for *E7* and “MT_hTERT” for *hTERT*) with untransfected parental cells (MT) were examined. Both lines exhibited proliferation similar to that of MT, with alkaline phosphatase activity and stemness-specific factor expression. They displayed differentiation potential into multi-lineage cells with no tumorigenic property. Overall, we successfully obtained HDDPC-derived immortalized cell lines using a PB-based transfection system. The resultant and parental cells were indistinguishable. Thus, *E7* and *hTERT* could immortalize HDDPCs without causing cancer-associated changes or altering phenotypic properties.

## 1. Introduction

Human deciduous teeth (HD) are predominantly discarded from children aged 6–12 years old, at which time they are replaced by adult teeth. However, following a report that stem cells could be isolated from human exfoliated deciduous teeth, dental pulp cells (DPCs) derived from HD (HDDPCs) became recognized as a useful resource for tissue (tooth) regeneration [1,2]. Toward this end, a thorough understanding of the properties of HDDPCs is required. However, almost all primary cultured HDDPCs are subject to the “Hayflick limit”, i.e., the number of times a normal human cell population will divide prior to cessation of cell division [3]. The length of telomeres located at both ends of chromosomes is known to regulate the “Hayflick limit”, with shortened telomere length being frequently associated with decreased cellular longevity and increased risk of cancer [4]. Notably, the inability of HDDPCs to proliferate for long periods in vitro often hampers innovative research on these cells. Therefore, the acquisition of HDDPCs that are capable of extended proliferation while retaining their other original properties is necessary.

It is currently possible to convert normal cells to those capable of proliferating in vitro without ceasing their cell division [5]. This event, termed “immortalization”, can be induced by transfection with simian virus 40 (SV40) or human papilloma virus 16 (HPV16) [6,7]. SV40-mediated immortalization is driven by the expression of SV40 large T antigen (TAg), which in turn inactivates key tumor suppressors including p53 and the retinoblastoma protein by directly binding to their proteins [8]. Notably, expression of the SV40 TAg, but not small T antigen, in human cells leads to the induction of chromosomal aberrations and polyploidy [9]. The E7 oncoprotein of HPV16 (HPV16-E7), which is recognized as an immortalizing agent in human epithelial cells [10], binds only to retinoblastoma family proteins, thereby preserving p53-induced apoptosis [11]. Moreover, unlike SV40 TAg, HPV16-E7 does not interact with the CBP/p300 transcriptional complex [12], suggesting that overexpression of HPV16-E7 in cultured human cells might preserve their own potential. Nevertheless, only a few reports exist regarding the successful immortalization of human cells using this viral oncogene [13].

Telomeres, which comprise the distal ends of human chromosomes, are composed of tandem repeats of a six-base sequence, TTAGGG, which is repeated up to approximately 10 kb in length [14]. Normal somatic cells shorten the telomere length by 50 to 100 bases with each cell division [15]. Then, when these somatic cells reach mortality stage I (M1) phase, the initial stage for a cell incapable of further cell proliferation, cell division is stopped [16]. Telomere end maintenance is effected via telomerase, a ribonucleoprotein polymerase that facilitates the addition of telomere repeats. Telomerase is composed of at least three components: human telomerase RNA (hTR), which contains the template sequence coding for telomere repeats; catalytic subunit human telomerase reverse transcriptase (hTERT), which possesses conserved reverse transcriptase motifs; and human telomerase-associated protein 1 (hTEP1), which exhibits similarity to the *Tetrahymena* telomerase protein p80 that is known to play a role in coordinating telomerase holoenzyme structure and/or recruiting telomerase regulatory factors [17,18,19]. Among these three major components, hTERT is considered to be of primary importance, as its expression is usually required for cell immortalization and long-term tumor growth [20,21]. hTERT is highly active in human germ-line and stem cells, and also in cancer and SV40-immortalized cells [22,23]. Notably, it was reported that undifferentiated dental pulp stem cells (DPSCs) express high telomerase activity, although this gradually decreased along with cell passage [24]. To date, successful immortalization driven by the expression of *hTERT* complementary DNA (cDNA) alone was reported in a variety of cell types related to the dental field, including human dental papilla cells [25], dental pulp cells [25,26,27], periodontal ligament cells [25,28], and gingival fibroblasts [25]. However, to our knowledge, no attempts were made toward the successful establishment of HDDPC-derived immortalized cells.

In our previous study [29], we established five primary HDDPC lines: MM, MT, SM, YY, and MA (#1–#5) [29]; the MT line used in the present study corresponds to cell line #5 of this previous work. This MT line was chosen since it was confirmed to express stem-cell-specific factors, as exemplified by octamer-binding transcription factor-3/4 (OCT3/4), and could be easily reprogrammed to form induced pluripotent stem (iPS) cells upon transfection with Yamanaka factors. However, the MT line exhibited lower proliferation rates and proliferation arrest after around passage 14 to 16. Immortalization of the MT cell line allowed us to use these cells as a “continuous cell line”, which will be beneficial for further characterization at the molecular level. Therefore, we chose this line in expectation that an immortalized MT cell line might be generated following transfection with a vector carrying *HPV16-E7* or *hTERT* cDNA. For efficient acquisition of immortalized MT cells, we employed the *piggy*Bac (PB) transposon-based gene delivery system, which was proven to be effective for obtaining transfectants from various cell types including human cells [30,31,32,33], goat fetal fibroblasts [34], porcine fetal fibroblasts [35], and mouse fibroblasts [36]. In the present study, we examined the properties of such established immortalized MT cells including cell proliferation rate, expression of stemness factors, anchorage-dependent growth, and non-tumorigenic potential in comparison with those of their parental cells in vitro and in vivo.

## 2. Results

### 2.1. Strategy for Generating Immortalized HDDPCs

We attempted to acquire immortalized cells by transfecting HDDPCs (MT line) with PB-based transposons containing immortalizing genes by using the Neon^®^ transfection system. This attempt was aimed to realize persistent (rather than transient) expression of immortalizing genes via their PB-mediated chromosomal integration [33]. The PB-based transposons and PB expression plasmid used in the present study are shown in Figure 1A. Briefly, the pTrans (pCX-mPB) vector [37] confers expression of the PB transposase gene under the control of *CAG*, a chicken β-actin-based systemic promoter [38]. pT-pac is a transposon vector carrying a puromycin acetyltransferase gene (*pac*) expression unit linked to the mouse phosphoglycerate kinase (*Pgk*) promoter [37]. pT-tdTomato is a transposon vector carrying tandem dimer Tomato (*tdTomato*) cDNA expression units (kindly provided by Dr. Roger Tsien) [35]. pT-EGFP is a transposon vector carrying an enhanced green fluorescent protein (*EGFP*) cDNA expression unit [37]. pT-E7 is a transposon vector carrying an *HPV16-E7* expression unit. pT-hTERT is a transposon vector carrying an *hTERT* cDNA expression unit.

The PB-based gene delivery system is known to be an efficient system to obtain stable HDDPC-derived transfectants [33]. As shown schematically in Figure 1B, the PB-based gene delivery system depends on transient expression of PB transposase, which binds to a PB acceptor site (also known as an inverted terminal repeat). In PB transposon vectors, a gene expression unit (GEU) is generally engineered to be surrounded by the PB in a plasmid backbone. When the GEU is integrated into a host chromosome, the PB transposase/PB acceptor complex removes the plasmid backbone and recognizes the TTAA consensus sequence present in chromosomal DNA to mediate integration of GEU alone into the host chromosome via TTAA [39,40].

### 2.2. Successful Establishment of Immortalized HDDPCs

Prior to transfection of HDDPCs with the listed transposons and pTrans, we addressed the issue of potential biased selection of certain types of HDDPCs when picking and propagating stable transfectants, as HDDPCs are considered to be composed of various cell types including multipotent stem cells. Consistent with this concept, we observed mosaic expression of alkaline phosphatase (ALP) in HDDPCs [29], suggesting that HDDPCs consist of a mixture of ALP-positive and -negative cells. Therefore, to avoid biased selection of stable transfectants, we collected all surviving colonies following transfection and subsequent transient selection with the selective drug (puromycin) through trypsinization, as described in Section 4.

We firstly assessed transfection efficiency through inspection for fluorescence in cells one day following transfection. Over 30% of cells exhibited fluorescence; moreover, the rate of fluorescence did not differ between cells transfected with pT-E7 + pT-tdTomato + pT-pac + pTrans and those with pT-hTERT + pT-EGFP + pT-pac + pTrans (a vs. c in Figure 1C). After selection in the presence of higher concentrations of puromycin (5 μg/mL) for one day, the cells were cultured in normal medium for more than 10 days until semi-confluency. Inspection of fluorescence in colonies surviving after puromcyin treatment (approximately 20 per dish) demonstrated that, in each group, >80% of colonies exhibited fluorescence (Figure 1Cb,d), although the strength of fluorescence appeared to differ slightly among colonies. The colonies in each dish were then trypsinized, pooled, and subjected to freezing for preparing cell stocks. The resultant lines were designated as MT_E7 (cells surviving after transfection with pT-E7 + pT-tdTomato + pT-pac + pTrans) and MT_hTERT (cells surviving after transfection with pT-hTERT + pT-EGFP + pT-pac + pTrans). These cells appeared to share the same morphology, showing fibroblastic shape similar to that of bone marrow-derived mesenchymal cells, and were indistinguishable from their parental MT line.

### 2.3. Characterization of Established MT_E7 and MT_hTERT Lines

#### 2.3.1. Transgenic Gene Expression and Cell Proliferation

We firstly assessed the presence of the introduced genes in the established MT_E7 and MT_hTERT lines by polymerase chain reaction (PCR) using genomic DNA. As shown in Figure 1D, MT_E7 yielded an amplicon corresponding to *HPV16-E7*. Similarly, MT_hTERT had a band corresponding to *hTERT* cDNA. Moreover, reverse-transcription (RT)-PCR demonstrated that the transcripts derived from the integrated *HPV16-E7* or *hTERT* expression unit were detectable (Figure 1E).

Both MT_E7 and MT_hTERT exhibited similar rates of cell proliferation as that of their parental MT line (Figure 2A). There were no significant differences in the number of cells among the three cell lines. Notably, proliferation in both lines appeared to reach a plateau on the seventh or ninth day of culture (Figure 2A), which was also consistent with the growth mode of the parental MT cell line.

#### 2.3.2. Stemness

Our previous work demonstrated that HDDPCs (including MT) exhibited stemness properties [29]. Here, we utilized cytological and molecular biological methods to examine whether both MT_E7 and MT_hTERT lines demonstrated similar properties to the parental MT line. We firstly examined ALP activity in MT, MT_E7, and MT_hTERT lines, as ALP expression is closely associated with undifferentiated immature cells such as somatic stem cells and embryonic stem cells/iPS cells [41,42]. Both MT_E7 and MT_hTERT lines were mosaically stained for ALP activity, as was the parental MT line (Figure 2B), suggesting a mixture of ALP-positive and -negative cells. Notably, this staining pattern for ALP activity was maintained even after passage 20 (data not shown). RT-PCR analysis demonstrated that both MT_E7 and MT_hTERT lines expressed mRNA for typical stemness factors as exemplified by octamer-binding transcription factor-3/4 (OCT-3/4), homeobox transcription factor (NANOG), and sex determining region Y-box 2 (SOX2) along with other differentiation-related proteins such as neuroectodermal stem cell-specific intermediate filament (NESTIN) and dentin sialophosphoprotein (DSPP) (Figure 2C). This pattern of expression was also observed in the parental MT line (Figure 2C). To confirm the results of RT-PCR, we next performed immunocytochemical staining using specific antibodies raised against OCT-3/4, SOX2, NANOG, and NESTIN (Figure 3). In our previous study, we found that HDDPCs including MT were enriched with undifferentiated cells [29]. Similarly, the present immunofluorescent staining revealed that MT_E7 and MT_hTERT lines, as well as MT lines, were positive for staining with the listed antibodies (Figure 3), suggesting no significant alteration in their stemness properties prior to and following immortalization (for MT_hTERT cells). In addition, immunostaining of DSPP, a differentiation marker of odontoblasts, was detected in all cell lines examined.

#### 2.3.3. Multipotency

We next examined the multipotent differentiation ability (at least, differentiation into neurogenic and osteoblastic lineages) of the immortalized HDDPCs. MT, MT_E7, and MT_hTERT lines were firstly cultured in neurogenic differentiation-inducing medium for one week and then subjected to Nissl staining. MT cells that were continuously cultured in normal medium were used as a negative control. In the experimental group that was cultivated in neurogenic differentiation-inducing medium, all of the cells tested exhibited neuronal morphology with elongated axons and dendrites (Figure 2Da,c), and Nissl bodies (arrows in Figure 2Da,c), a marker of neuronal cells, evident around the nucleus. Untreated control MT cells did not show such alteration (Figure 2Dd). Next, MT, MT_E7, and MT_hTERT lines were cultured in osteogenic differentiation-inducing medium for two weeks and then subjected to Alizarin red staining. All cells exhibited heavily calcified deposits (Figure 2De,g). In contrast, MT cells cultured continuously in normal medium remained intact (Figure 2Dh).

#### 2.3.4. Immortalization

Generally, immortalized cell lines continue to grow and divide indefinitely in vitro under optimal culture conditions, which resembles the behavior of in vitro cultivated tumors cells, although immortalized cells exhibit many properties indistinguishable from somatic cells existing in a multicellular organism [43,44]. In particular, anchorage-independent growth of cancer cells constitutes a key aspect of the tumor phenotype [45]. We examined this property in MT, MT_E7, and MT_hTRET cell lines by seeding each onto 2% agar-coated dishes, as shown schematically in the left panel of Figure 4A. P19 cells, a murine teratocarcinoma cell line [46], were also seeded as a positive control. At two weeks after seeding, all cells were observed to float while forming small cell aggregates (Figure 4Aa,d). Notably, the cell aggregates from MT, MT_E7, and MT_hTRET lines appeared to be dead, whereas those from P19 cells appeared to be still viable. These floating aggregates were collected, re-seeded onto a gelatin-coated 60-mm dish, and cultivated for one additional week to allow them to attach and spread out onto the dish surface (Figure 4A, left). MT and its derivative cells (MT_E7 and MT_hTRET) both failed to attach and proliferate on gelatin-coated dishes (Figure 4Ae,g), whereas P19 cells exhibited attachment to the dish and better proliferation (Figure 4Ah).

#### 2.3.5. Tumorigenic potential

We also examined whether the immortalized HDDPCs could exhibit tumorigenic properties when they were transplanted into the internal area of the pancreatic parenchyma of nude mice as an in vivo tumorigenic assay (Figure 4B, left). Concomitantly, we transplanted P19 cells as a positive control. At six weeks after transplantation, generation of a distinct solid tumor was visible in the P19 group (Figure 4Ba, arrows). In contrast, nude mice transplanted with MT_E7 cells did not show any distinct solid tumor in their skin surface when assessed at eight weeks following transplantation (Figure 4Bb, arrowheads). Visual inspection of the pancreas dissected from the MT_E7 grafted mice revealed no distinct tumor formation (Figure 4Bc) in the grafted area of the pancreas, although expression of tdTomato-derived red fluorescence could be observed (Figure 4Bd).

## 3. Discussion

In this study, we successfully immortalized HDDPCs through transfection with PB-based transposons carrying a GEU for *HPV16-E7* or *hTERT* cDNA. The established HDDPCs exhibited cell proliferation rates similar to those of their parental cells (Figure 2A), along with ALP activity (Figure 2B) as evaluated by cytochemical staining. Notably, undifferentiated mesenchymal cells included in the DP are reported to demonstrate secondary dentin-forming and potential calcification abilities [47,48]. They also exhibit ALP activity, which is thought to be an indicator of calcification [49]. However, we consider that ALP may also serve as to indicate a state in which DP cells are enriched with immature stem cells, as HDDPCs enriched for both ALP and OCT-3/4 expression are more easily reprogrammed to generate iPS cells upon forced expression of reprogramming factors [29,50]. Notably, Cristofalo et al. demonstrated that human fibroblasts exhibit ALP activity at the initial stage of their isolation, whereas this tends to be lost along with cell aging [51]. Furthermore, Rikitake et al. demonstrated ALP activity in immortalized human DP cells along with their parental cells [52]. In the present study, the ALP activity in the primary isolated cells decreased as passage number increased, with activity eventually disappearing almost completely. In contrast, although the immortalized cells cultured for 10 passages demonstrated an approximately 40% reduction in ALP activity compared to that in the initial stage of cells, this level was maintained over 150 passages. In this context, ALP may represent a good marker for indicating the state of mesenchymal cells that continue to grow and divide indefinitely in vitro.

The possible occurrence of phenotypic alteration in the immortalized cells following transfection with immortalizing factors is a main issue to be considered when researchers intend to create immortalized cell lines from primary cultured cells. Our presently established lines MT_E7 and MT_hTERT were indistinguishable from their parental MT cells with regard to cell proliferation rate, cell morphology, expression of stemness-specific genes, and potential to differentiate into multi-lineage cells (Figure 2 and Figure 3). Notably, Galler et al. demonstrated that SV40 TAg-immortalized human DPCs retained many of the phenotypic characteristics observed in their parental cells [53]. In addition to dental cells, similar findings were also observed when immortalizing genes were delivered to various cell types such as human corneal epithelial cells [54], human osteoblastic cells [55], a human airway epithelium derived basal cell line [56], human adipose-derived stem cells [22], and human prostate epithelial cells [57]. Furthermore, similar to the cells in the present study, which exhibited multiple differentiation capability upon differentiation induction (Figure 2D), other stem cell-like cells also still exhibited such capability even following immortalization [22,56,57,58].

Another aspect related to immortalization would be the occasional conversion to cells with high tumorigenic ability. It is generally considered that cancer cells with metastatic potential have a high telomerase activity [59,60]. Furthermore, as with SV40 TAg, the HPV16-E7 protein was found to induce genomic instability such as centrosome abnormalities [61]. Therefore, we assessed this possibility using in vitro anchorage-dependent growth and in vivo tumorigenic assays. In the former case, extracellular matrix proteins or the surface of a tissue-culture dish can serve as an anchor for the growth of almost all cells other than hematopoietic cells. In the absence of anchors, anchor-dependent normal cells begin to limit their proliferation, whereas cancer cells generally survive and grow even under this condition [45]. In the present study, we employed 2% agarose onto which normal (but not cancer) cells cannot attach and survive owing to the absence of anchors. Accordingly, immortalized HDDPCs (but not P19 cells) failed to survive under this condition (Figure 4A), suggesting that they retain the property of an anchorage-dependent cell line. In the latter case, cancer cells grafted into immunodeficient mice such as nude mice generally form solid tumors. As expected, P19 (but not immortalized HDDPCs) cells formed solid tumors in nude mice (Figure 4B, a vs. b), suggesting that our present immortalized HDDPCs do not exhibit cancer-associated changes. This is consistent with the results of several previous reports wherein immortalized cells generally failed to show tumorigenic property when cultivated on agar-coated dishes and/or inoculated underneath the skin of immunocompromised mice [62,63,64,65,66].

In this study, we employed a PB-based transposon system for efficient acquisition of HDDPCs transfected with immortalized genes such as *HPV16-E7* and *hTERT*, as the PB system was shown to be useful for acquiring several stable HDDPC lines [33]. We obtained over 30 colonies of MT cells following single co-transfection of HDDPCs (5 × 10^4^ cells) with PB transposons (carrying fluorescent marker genes or a drug resistance gene) and a PB transposase expression vector, and subsequent transient (for one day) selection with higher concentration (5 μg/mL) of puromycin, which can confer efficient acquisition of stable transfectants. The resulting colonies were then collected together and successfully maintained in normal medium without losing their fluorescence. Thus, the PB-based transfection system appears to be suitable for efficient acquisition of immortalized cells from hard-to-transfect cells, such as primary cells (i.e., HDDPCs). Notably, to our knowledge, there were few previous reports showing successful isolation of immortalized cells by the transposon-based transfection approach [67].

DPSCs are currently being considered as useful resources for research in regenerative medicine, as they exhibit high proliferative ability and the ability to differentiate into several types of cells including osteoblasts, adipocytes, chondrocytes, and nerve cells [68,69]. Our present immortalized HDDPCs share multiple properties with their parental cells including similar growth rate, expression of stemness-specific factors, and lack of tumorigenic potential. These properties would be beneficial for basic research such as gene expression analysis, identification of stem cells showing lineage-specific differentiation, and exploration of gene function using genetic engineering technology to optimize strategies and protocols prior to clinical application of HDDPCs.

## 4. Materials and Methods

### 4.1. Mice

Immunodeficient female mice (BALB/c-nu/nu; 8–20 weeks old) were purchased from CLEA Japan Ltd. (Tokyo, Japan). Mice were kept under a 12-h light/12-h dark schedule (lights on from 7:00 a.m. to 7:00 p.m.) and allowed food and water ad libitum. All animal experiments were performed in agreement with the guidelines of Kagoshima University Committee on Recombinant DNA Security and approved by the Animal Care and Experimentation Committee of Kagoshima University (permit no. 25,035 and 25036; dated 8 August 2013 permission no. 16008; valid from 9 August 2016 to 31 March 2019). All surgeries were performed following intraperitoneal (IP) injection of three anesthetics (medetomidine (0.75 mg/kg; Nippon Zenyaku Kogyo Co., Ltd., Fukushima, Japan), midazolam (4 mg/kg; Sandoz K.K., Tokyo, Japan), and butorphanol (5 mg/kg; Meiji Seika Pharma Co., Ltd., Tokyo, Japan)). Consciousness was restored by subcutaneous injection of atipamezole (3.75 mg/kg; Nippon Zenyaku Kogyo Co., Ltd.), an antagonist of medetomidine; then, mice were warmed using an electric plate warmer. All efforts were made to minimize suffering.

### 4.2. Cells

The DPCs were collected from the healthy deciduous teeth that were extracted owing to replacement failure in our department. HDDPCs were obtained with informed patient consent, and the protocols used in this study were approved by the Ethics Committee of Kagoshima University Graduate School of Medical and Dental Sciences (permission no. 27-11; valid from 29 May 2015 to 31 March 2020). HDDPCs (MT line) previously isolated in-house [29] were maintained in 60-mm gelatin-coated dishes (#4010-020; Iwaki Glass Co., Ltd., Tokyo, Japan) containing Dulbecco’s modified Eagle’s medium (DMEM) (#11995-081; Invitrogen Co., Carlsbad, CA, USA) with 20% heat-inactivated fetal bovine serum (FBS) (#SFMB30-2239; Equitech Bio Inc., Kerrville, TX, USA), 50 U/mL penicillin, and 50 mg/mL streptomycin (#15140-122; Invitrogen) (DMEM/20% FBS) at 37 °C in an atmosphere of 5% CO_2_ in air. Medium was exchanged every three days. After 3–4 passages initiating from primary cultivation, the MT line was frozen, yielding up to 10 or more frozen stocks. On around passage six, the MT line was used for transfection experiments. P19 cells were used for anchorage-dependent growth and in vivo tumorigenesis assays, as mentioned below. They were grown in DMEM supplemented with 10% heat-inactivated FBS, 50 U/mL penicillin, and 50 mg/mL streptomycin (DMEM/10% FBS).

### 4.3. Construction of PB Transposon Vectors

The PB-related expression vectors shown in Figure 1A were obtained from others or generated in-house using standard cloning procedures. For construction of pT-E7, a 477-bp fragment containing *HPV16-E7* was firstly isolated by PCR using genomic DNA from HeLa cells (which are known to contain HPV18 in their genome) [70]. The primers used were 5′-*Eco*RI-primer (sense): 5′–GGA ATT CGC CAC CAT GGA CTA CAA GGA TGA CGA TGA CAA GAT GTA TGG ACC TAA GGC AAC ATT–3′, containing an *Eco*RI site and FLAG sequence in front of *HPV16-E7*, and 3′-NotI-*Eco*RI-primer (reverse): 5′–GGA ATT CGC GGC CGC GAT ATT ACA TCT CCT GTT T–3′, containing an *Eco*RI site. The isolated fragment was then inserted into the *Eco*RI site present in the third exon of the rabbit β-globin gene in pCAGGS [38]. The fragment containing the CAG-based E7 expression unit was next inserted into the PB-based vector pPB-MCS-P5 to create pT-E7 [35]. For construction of pT-hTERT, an approximately 4.9-kb *Bgl* II/*Cla* I fragment containing cytomegalovirus enhancer/promoter, *hTERT* (*hEST2*) cDNA, and the *SV40* late poly(A) site was firstly isolated from pCIneo-hEST2-HA (#1782; Addgene, Cambridge, MA, USA) and then inserted into pPB-MCS-P5. All plasmids were grown in *Eshcerichia coli* DH5α and purified using the MACHEREY-NAGEL plasmid purification kit as described by Sato et al. [71].

### 4.4. Generation of Immortalized HDDPC Lines

For transfection with PB-based transposon vectors, MT cells (5 × 10^4^) were electroporated in a solution (100 µL) of R-buffer (Invitrogen) containing pT-E7 (1 μg) + pT-pac (1 μg) + pT-tdTomato (1 μg) + pTrans (1 μg) or pT-hTERT (1 μg) + pT-pac (1 μg) + pT-EGFP (1 μg) + pTrans (1 μg) using a Neon^®^ Transfection System (#MPK10096; Invitrogen) under the recommended electroporation condition #4 (1600 V, 20 ms, 1 pulse). Electroporated cells were then seeded onto a gelatin-coated 60-mm dish containing antibiotic-free DMEM/20% FBS. Six days after transfection, cells were cultured in DMEM/20% FBS containing 5 μg/mL puromycin (#ant-pr-1; InvivoGen Inc., San Diego, CA, USA) for one day. Then, the medium was changed to normal medium (DMEM/20% FBS) and the cells were cultured for more than 10 days until confluency. Cells were then trypsinized and further propagated. A portion of cells were subjected to cell freezing. The established cells were designated “MT_E7” (obtained after transfection with an *E7* expression transposon) and “MT_hTERT” (obtained after transfection with an *hTERT* expression transposon).

### 4.5. Fluorescence Observation

Fluorescence in the cells was examined using a BX60 fluorescence microscope (Olympus, Tokyo, Japan) with DM505 (BP460-490 and BA510IF; Olympus) and DM600 filters (BP545-580 and BA6101F; Olympus), which were utilized for *EGFP*-derived green fluorescence and *tdTomato*-derived red fluorescence, respectively. Microphotographs were taken using a digital camera (FUJIX HC-300/OL; Fuji Film, Tokyo, Japan) attached to the fluorescence microscope and printed using a Mitsubishi digital color printer (CP700DSA; Mitsubishi, Tokyo, Japan).

### 4.6. Cell Growth Assay

To evaluate the cell proliferation rate of each line (MT, MT_E7, and MT_hTERT), cells (1 × 10^4^) were seeded into wells of a gelatin-coated 24-well plate (#3820-024; Iwaki Glass Co., Ltd.) containing DMEM/20% FBS. The cells were collected by trypsinization on days one, three, five, seven, or nine after seeding, and then the cell number was counted using a disposable hemocytometer (#521-10; Funakoshi, Tokyo, Japan). In this case, we observed cell confluency on day nine after cell seeding; therefore, we only counted the cell number up to day nine of culture. At least three wells per line were examined and the average cell number was plotted. Differences in cell proliferation among groups were evaluated using the Kruskal–Wallis test with a significance level of 0.05.

### 4.7. ALP Assay

Cells were cultivated in a gelatin-coated 24-well plate containing DMEM/20% FBS until reaching 80–90% confluency. They were then fixed using 4% paraformaldehyde (PFA) for 5 min at room temperature (RT) (approximately 25 °C) and washed three times with phosphate-buffered saline without Ca^2+^ and Mg^2+^ (PBS(−)) prior to cytochemical staining for ALP activity using the Leukocyte Alkaline Phosphatase kit (#ALP-TK1; Sigma-Aldrich, Tokyo, Japan). The ALP activity can be visualized by the appearance of red brown products upon ALP-mediated conversion of α-naphtholum coupled with diazonium salt.

### 4.8. PCR Analysis

Genomic DNA from HDDPCs was isolated as previously described [72]. PCR was performed in a total volume of 10 μL containing 10 mM Tris-HCl (pH 8.3), 50 mM KCl, 1.5 mM MgCl_2_, 0.25 mM of each dNTP (dATP, dTTP, dGTP and dCTP), 1 mM of each primer (forward and reverse), 2 μL of genomic DNA (approximately 5 ng), and 0.5 units of *rTaq* polymerase (#R001; TaKaRa Shuzo Co., Ltd., Tokyo, Japan). For detection of pT-E7, primer set β-gl-1 (5′–CTC CTG GGC AAC GTG CTG GT–3′; Figure 1A) (accession no. V00882) and E7-RV (5′–TGG CTT CAC ACT TAC AAC ACA–3′; Figure 1A) (accession no. MH196518) was used, which yields a 293-bp product from the upper region of *E7*. Similarly, for detection of pT-hTERT, primer set hTERT-3S (5′–TTT CTG GAT TTG CAG GTG AAC–3′; Figure 1A) and hTERT-3RV (5′–GTG ATG TCC AGC TTG GTG TCC–3′; Figure 1A) was used, which yields a 290-bp product from the *hTERT* cDNA (accession no. AB085628). For detection of pT-pac, primer set puro-2S (5′–TCT ACG AGC GGC TCG GCT TCA–3′; Figure 1A) and puro-4RV (5′–TCA GGC ACC GGG CTT GCG GGT–3′; Figure 1A) was used, which yields a 92-bp product from *pac* [35]. As a control, primer set GAPDH-5′-S (5′–GCC TCA AGA CCT TGG GCT GGG–3′) and GAPDH-3′-S (5′–TGG CGA CGC AAA AGA AGA TGC–3′) was used for amplification of the endogenous glyceraldehyde-3-phosphate dehydrogenase (*GAPDH*) gene (accession no. NM_002046) as a 164-bp product. As a negative control, genomic DNA from untransfected HDDPCs was used. As a positive control, 5 ng of each plasmid listed in Figure 1A was used. PCR was performed with 40 cycles of 96 °C for 10 s, 56 °C for 1 min, and 72 °C for 2 min. The PCR products (5 µL) were separated on a 2% agarose gel and visualized using ethidium bromide.

### 4.9. RT-PCR Analysis

To measure the expression of endogenous *OCT-3/4*, *SOX2*, *NANOG*, *NESTIN*, *DSPP*, and *GAPDH* mRNA by RT-PCR, the cells were cultivated in 60-mm gelatin-coated dishes containing DMEM supplemented with 20% FBS until reaching 90% confluency. The cells were harvested via standard trypsinization and subjected to lysis for RNA preparation. Total RNA (approximately 4 μg) was isolated from HDDPCs using the RNeasy Mini kit (#74104; QIAGEN, Hilden, Germany), and then reverse-transcribed using a Super Script III First-Strand Synthesis System (#18080051; Thermo Fisher Scientific, Inc., Waltham, MA, USA), according to the manufacturer’s protocol. Briefly, RNA (up to 5 μg) was mixed with 1 μL of 50 μg/μL random hexamer primer and 1 μL of 10 mM dNTPs in a 1.5-mL microtube, then treated at 65 °C for 5 min. Next, 10 μL of cDNA synthesis mix (comprising 10× RT buffer, 25 mM MgCl_2_, 0.1 M dithiothreitol (DTT), RNase OUT, and Super Script III RT) was added to the tube to allow reverse transcription under the conditions of 25 °C for 10 min, 50 °C for 50 min, and 85 °C for 5 min. The residual RNA was digested by addition of 1 μL of RNase and subsequent incubation at 37 °C for 20 min. The undiluted cDNA samples (1 µL) were then amplified by PCR in a total volume of 20 μL using AmpliTaq Gold^®^ 360 Master Mix (#4398881; Applied Biosystems, Foster City, CA, USA). PCR was performed with 37 cycles of denaturation at 95 °C for 60 s, annealing at 62 °C for 60 s, and extension at 72 °C for 60 s in a PC708 thermal cycler (Astec, Fukuoka, Japan). A negative, no-template control (designated −RT) was included for each reaction. PCR primers were based on those used in our pervious paper [29]. The products (5 µL) were analyzed by 2% agarose gel electrophoresis and visualized with GelRed Nucleic Acid Gel Stain (#41002; Biotium, Inc., Fremont, CA, USA).

For detection of expression of the introduced immortalized gens, the undiluted cDNA samples (1 μL) prepared from the immortalized cells were subjected to the first PCR in a total volume of 20 μL using the AmpliTaq Gold^®^ 360 Master Mix with primer sets βA-1 (5′–TCT GAC TGA CCG CGT TAC TCC CAC A–3′; Figure 1A; accession no. X00182)/E7-RV for *HPV16-E7* mRNA and hTERT-2S (5′–GGG GTC TTG CGG CTG AAG TGT–3′; Figure 1A; accession no. AB085628)/hTERT-2RV (5′–GAG TGG CAC GTA GGT GAC ACG–3′; Figure 1A; accession no. AB085628) for *hTERT* mRNA. GAPDH Forward and GAPDH Reverse were also used for amplification of endogenous *GAPDH* mRNA [29]. The PCR condition was the same as mentioned above. In this case, untransfected HDDPC-derived RNA was used as the negative control. The resulting products (1 μL; only for MT_E7 and MT_hTERT) were then subjected to nested PCR using primer set β-gl-1S (5′–GTT GTG CTG TCT CAT CAT TTT–3′; accession no. V00882)/E7-2RV (5′–AAT GTT GAT GAT TAA CTC CAT–3′; accession no. MH196518) for *HPV16-E7* mRNA and hTERT-3S/hTERT-3RV for *hTERT* mRNA in a total volume of 20 μL using AmpliTaq Gold^®^ 360 Master Mix. PCR with β-gl-1S/E7-2RV, hTERT-3S/hTERT-3RV, and GAPDH Forward/GAPDH Reverse would theoretically yield 212-, 290-, and 325-bp amplicons, respectively. The products (2 μL) were analyzed by 2% agarose gel electrophoresis and visualized with GelRed Nucleic Acid Gel Stain.

### 4.10. Immunocytochemical Staining

Cells were seeded onto a gelatin-coated 24-well plate containing DMEM/20% FBS. Cells at 60–70% confluency were fixed with 4% PFA and washed three times with PBS(−). They were then subjected to permeabilization via incubation in 0.1% Triton X-100 (#T8787; Sigma-Aldrich, St. Louis, MO, USA) in PBS(−) (*w*/*v*) for 3 min at RT. After washing with PBS(−) containing 1% normal goat serum (NGS) (Invitrogen) (PBS/NGS), cells were subjected to blocking via incubation in 20% AquaBlock tm/EIA/WB (#PP82; East Coast Biologics, Inc., North Berwick, USA) for 30 min at 4 °C. After washing with PBS/NGS, cells were stained with primary antibodies against OCT3/4 (#sc-9081; clone H-134, 1:200; Santa Cruz Biotechnology, Dallas, TX, USA), SOX2 (#SAB2701974; 1:200; Sigma-Aldrich), NANOG (#RCAB0004P; 1:200; Repro Cell, Kanagawa, Japan), NESTIN (#N5413; 1:200; Sigma-Aldrich), and DSPP (#HPA036230; 1:200; Sigma-Aldrich) overnight at 4 °C. After washing with PBS/NGS, cells were then reacted with the secondary antibodies, fluorescein isothiocyanate (FITC)-conjugated goat anti-mouse immunoglobulin G (IgG) γ chain antibody (1:200; Millipore-Chemicon, Darmstadt, Germany), Alexa Fluor 647-conjugated anti-rabbit IgG F(ab’)2 fragment (1:200; Cell Signaling Technology, Tokyo, Japan), or Alexa Fluor 488-conjugated goat anti-rabbit IgG H&L (1:200; Abcam, Cambridge, UK) for approximately 2 h at 4 °C. After washing with PBS/NGS, nuclear staining was performed using 4′,6-diamidino-2-phenylindole (DAPI) (#H-1200; Vector Laboratories Inc., Burlingame, CA, USA) for 10 min at RT. Fluorescence was examined under a BX60 fluorescence microscope.

### 4.11. Induction of In Vitro Differentiation

Cells were seeded onto a gelatin-coated 24-well plate containing DMEM/20% FBS and cultured until 80–90% confluency. Thereafter, the medium was changed to differentiation-inducing medium, as described below. For inducing neurogenic differentiation, cells were cultured in mesenchymal stem cell neurogenic differentiation medium (#C-28015; PromoCell, Heidelberg, Germany). After one week, the cells were fixed with 4% PFA and incubated with 0.1% cresyl violet solution (#038-0482; Wako Pure Chemical Industries, Ltd., Osaka, Japan) for 30 min at RT to stain the cytoplasm of neurons with Nissl substance. For inducing osteogenic differentiation, cells were cultured in osteogenic differentiation medium (#KBDSTC103; DS Phama, Osaka, Japan) for approximately 14 days. Then, the cells were fixed with 4% PFA and stained with Alizarin Red S (#ARD-A1; PG Research, Tokyo, Japan) for 30 min at RT. Cells grown in normal growth medium in the remaining wells were used as a negative control.

### 4.12. Anchorage-Dependent Growth Assay

HDDPC-derived cells (MT, MT_E7, and MT_hTERT) (approximately 0.5 × 10^5^) or P19 cells (approximately 5 × 10^3^) were seeded onto a 2% agarose (#50013R; ME-agarose, Iwaikagaku Co., Tokyo, Japan)-coated 60-mm dish with DMEM/20% FBS (or DMEM/10% FBS for P19 cells) and cultured for two weeks at 37 °C in an atmosphere of 5% CO_2_ in air, as described in the left panel of Figure 4A. Thereafter, the cells were seeded onto a gelatin-coated 60-mm dish with DMEM/20% FBS (or DMEM/10% FBS for P19 cells) and cultured for one week at 37 °C in an atmosphere of 5% CO_2_ in air.

### 4.13. In Vivo Tumorigenic Assay

For inducing solid tumor formation in vivo, MT_E7 (approximately 1 × 10^4^; expressing tdTomato) or P19 cells (approximately 1 × 10^3^; used as positive control) were injected into an internal area of the pancreatic parenchyma of nude mice (BALB/cAJcl-*nu/nu*; CLEA Japan Ltd., Tokyo, Japan), as described in the left panel of Figure 4B. With this method, it is possible to induce solid tumor formation starting from a small number (approximately 10^3^) of cells such as iPS cells [73]. Briefly, cells (dissolved in approximately 3 μL of culture medium) were aspirated using a glass micropipette connected to the mouthpiece. Under anesthesia, the spleen and pancreas of mice were pulled out, and approximately 1 μL of the solution was injected by inserting the micropipette into the pancreatic parenchyma under a dissecting microscope. Injections were performed toward a total of three different pancreatic sites. At six to eight weeks post-transplantation, mice were inspected for the presence/absence of solid tumors in the injected area of the pancreas.

## 5. Conclusions

We demonstrated that the PB-based gene delivery system enables the generation of immortalized cells from HDDPCs via transfection with transposon carrying *HPV16-E7* or *hTERT* cDNA. The resultant immortalized cells exhibited properties indistinguishable from those of their parent cells with no cancer-associated changes. These properties would be beneficial for basic research such as exploration of gene function using genetic engineering technology to optimize strategies and protocols prior to the clinical application of HDDPCs.

## Figures and Tables

**Figure 1 ijms-20-04904-f001:**
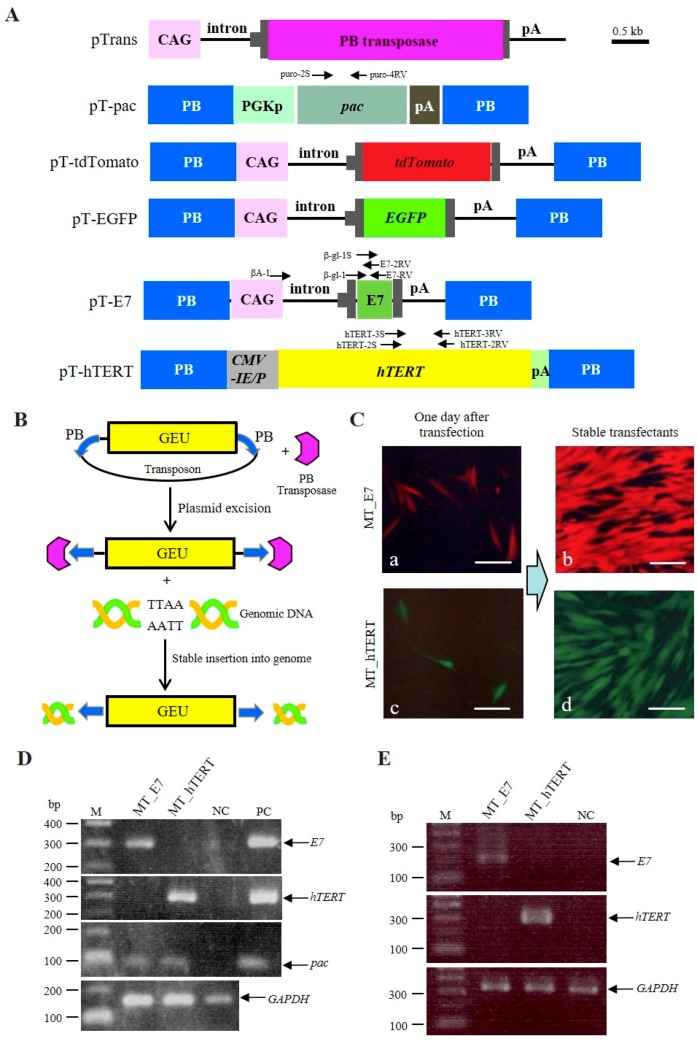
Establishment of immortalized human deciduous tooth-derived dental pulp cells (HDDPCs). (**A**) Plasmid vectors used for generating immortalized HDDPCs. The location of each primer is denoted above the construct. CAG, cytomegalovirus enhancer + chicken β-actin promoter; CMV-IE/P, cytomegalovirus enhancer and promoter; pA, poly(A) sites; *pac*, puromycin resistance gene; *EGFP*, enhanced green fluorescent protein complementary DNA (cDNA); *tdTomato*, tandem dimer Tomato cDNA; *E7*, *E7* of human papilloma virus 16 (HPV16); *hTERT,* human telomerase reverse transcriptase cDNA; *PGKp*, mouse phosphoglycerate kinase promoter; PB, acceptor site in the *piggy*Bac system; *PB transposase*, PB transposase gene. (**B**) Mechanism of target gene transfer to a host chromosome by the PB system. A gene expression unit (GEU) is engineered to be surrounded by the PB in a plasmid backbone. When the GEU is integrated into a host chromosome, the PB transposase/PB acceptor complex removes the plasmid backbone and recognizes the TTAA consensus sequence present in chromosomal DNA to mediate integration of GEU alone into host chromosome via TTAA. (**C**) Fluorescence in HDDPCs after transfection with pT-E7 + pT-tdTomato + pT-pac + pTrans (**a**,**b**) or pT-hTERT + pT-EGFP + pT-pac + pTrans (**c**,**d**). When fluorescence was inspected one day after transfection, over 30% of the cells (as shown in (**a**,**c**)) were successfully transfected. Almost all cells were also fluorescent (as shown in (**b**,**d**)) when the colonies generated after transient selection with puromycin were collected and propagated. Bar = 10 μm. (**D**) PCR analysis of genomic DNA isolated from the established immortalized HDDPCs (MT_E7 and MT_hTERT). Simultaneously, the same amounts of DNA were PCR-amplified using primers for detection of the endogenous *GAPDH* gene and used as internal controls. MT_E7 yielded a band corresponding to *E7.* Similarly, MT_hTERT exhibited a band corresponding to *hTERT* cDNA. Both lines produced a band corresponding to *pac*. M, 100-bp ladder markers; NC, genomic DNA from intact MT used as a negative control; PC, plasmids (5 ng) used as positive controls for each target gene/cDNA. (**E**) RT-PCR analysis of messenger RNA (mRNA) coding for *E7*, *hTERT,* and *GAPDH* in MT_E7 and MT_hTERT lines. NC, mRNA from intact MT used as a negative control. M, 100-bp ladder markers.

**Figure 2 ijms-20-04904-f002:**
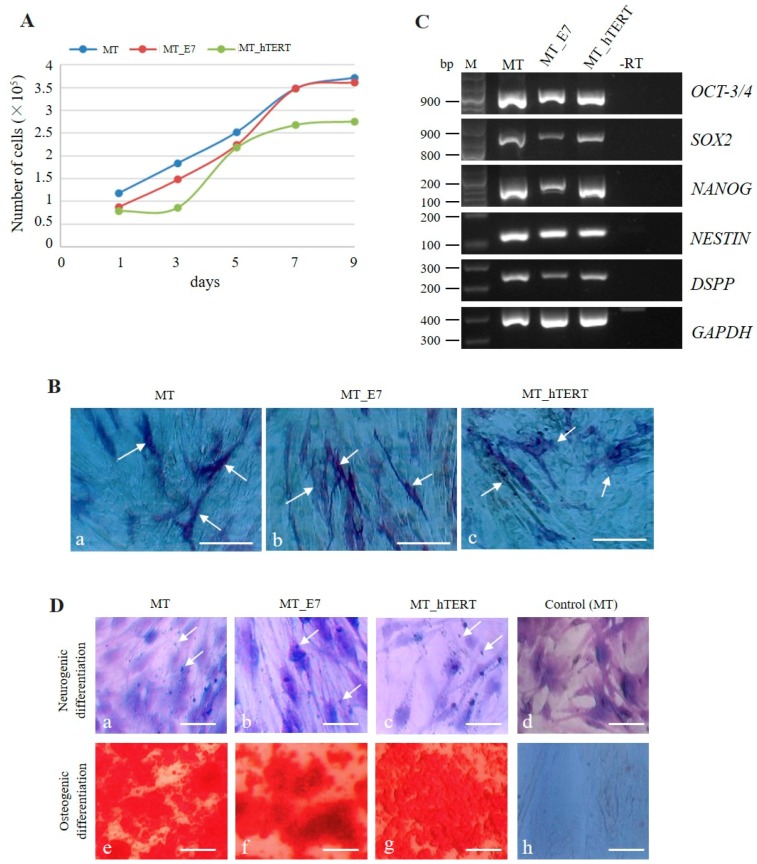
Characterization of the established MT_hTERT and MT_E7 lines. (**A**) Proliferation curves for parental MT, and MT_E7 and MT_hTERT lines. After seeding onto a 48-well plate, the cells were harvested from one to nine days after seeding, and the cell number was counted and plotted. (**B**) Cytochemical staining of MT, MT_E7, and MT_hTERT cells for alkaline phosphatase (ALP) activity. All cell lines exhibited positive staining for ALP (shown by arrows). Bar = 10 μm. (**C**) RT-PCR analysis of MT, MT_E7, and MT_hTERT lines using primer sets as shown in our previous paper (29). M, 100-bp ladder markers; −RT, PCR performed in the absence of reverse transcription products. (**D**) Testing for multipotent differentiation ability of MT (**a**,**e**), MT_E7 (**b**,**f**), and MT_hTERT (**c**,**g**) lines. Nissl staining was performed after incubation in neurogenic medium for one week (**a**–**c**) or MT incubated in normal medium (**d**). Nissl bodies (arrows) were discernible around the nucleus in the cells treated with neurogenic medium (**a**–**c**) but not in MT cells incubated in normal medium (**d**). Alizarin Red S staining was performed after incubation in osteogenic medium for two weeks (**e**–**g**) or MT incubated in normal medium (**h**). All of the cells treated with osteogenic medium exhibited heavily calcified deposits (**e**–**g**), which were not observed in MT cells incubated in normal medium (**d**). Bar = 10 μm.

**Figure 3 ijms-20-04904-f003:**
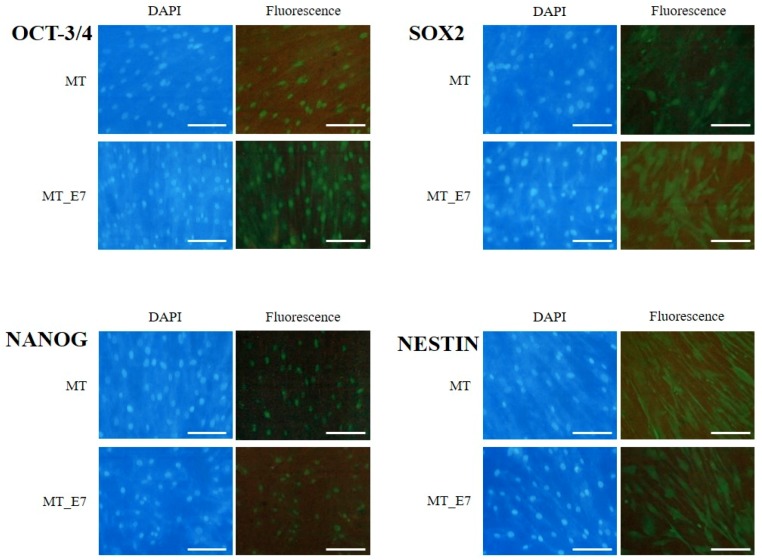
Immunocytochemistry of MT and MT_E7 lines using antibodies raised against octamer-binding transcription factor-3/4 (OCT-3/4), sex determining region Y-box 2 (SOX2), homeobox transcription factor (NANOG), and neuroectodermal stem cell-specific intermediate filament (NESTIN). After immunostaining with antibodies, cells were counterstained with 4’,6-diamidino-2-phenylindole (DAPI) to visualize the location of nuclei. Bar = 100 μm.

**Figure 4 ijms-20-04904-f004:**
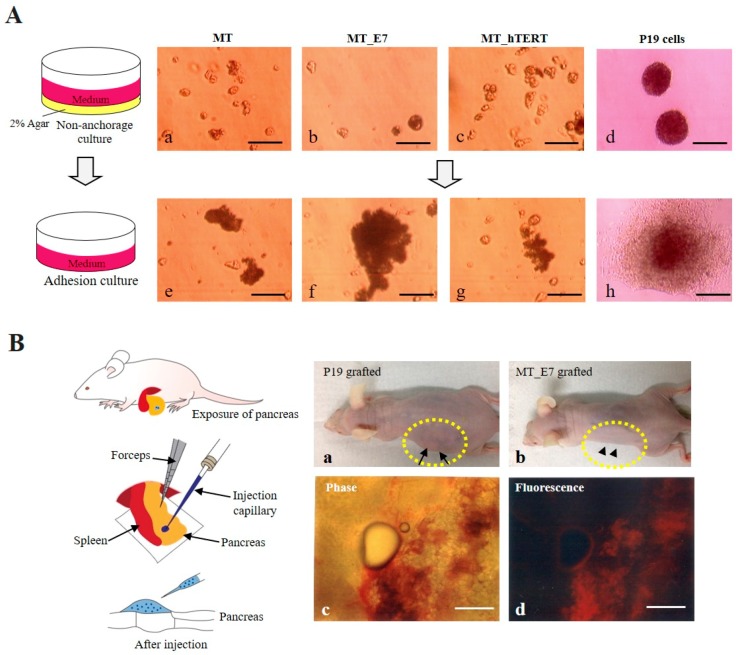
In vitro (**A**) and in vivo (**B**) tumorigenic assays. (**A**) Anchorage-dependent growth of MT, MT_E7, and MT_hTERT lines. Cells (including P19 cells) were seeded onto a 2% agar-coated dish, as shown in the left panel. After two weeks, floating cells along with small cell aggregates were observed for all lines (**a**–**d**). These floating cells were collected and then re-seeded onto a gelatin-coated 60-mm dish, as shown in the left panel. One week after re-seeding, MT, MT_hTRET, and MT_E7 lines failed to attach and spread out onto the surface of a dish (**e**–**g**), whereas P19 cells attached to the dish and proliferated well (**h**). Bar = 100 μm. (**B**) In vivo tumor formation of MT_E7 cells. Cells (MT_E7 or P19) were injected into an internal area of the pancreas of nude mice under a dissecting microscope, as shown in the left panel. Six weeks after transplantation, generation of distinct solid tumor was discernible in the P19-bearing mice (indicated by arrows enclosed by the dotted lines in (**a**)). However, nude mice transplanted with MT_E7 did not show any signs of distinct solid tumor formation (indicated by arrowheads enclosed by the dotted lines in (**b**)). When the transplanted region of the pancreas was dissected and inspected, no distinct tumor formation was observed (**c**), although tdTomato-derived red fluorescence was evident (**d**). Bar = 100 μm.

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
