# Peer review of "piggyBac Transposon-Based Immortalization of Human Deciduous Tooth Dental Pulp Cells with Multipotency and Non-Tumorigenic Potential"

_ijms, 2019, doi:10.3390/ijms20194904_

Round 1

Reviewer 1 Report

In this study, the authors have developed a piggyback transposon-based immortalization of human deciduous tooth dental pulp cells. The immortalized cells are shown to have multipotency and non-tumorigenic potential. Generally the results are interesting. Some revisions are suggested to improve the manuscript.

Page 6: Both MT E7 and MT hTERT exhibited similar rates of cell proliferation…… (Fig. 2A). Any analysis of data of 3 strains in Fig. 2A? MT_hTERT seems lower than MT and MT_E7? For Fig. 2B: Whether MT_hTERT cells are also shown less ALP activities? For Fig. 2C: Whether DSPP mRNA expression (a potential expression marker of odontoblasts) is easily identified by RT-PCR? What is the culture condition of cells (medium or mineralized medium?  For how many days?) Page 8: ….exhibited neuronal morphology with elongated axons and dendrites (Fig. 2D) – easily discernable and specific? Page 7, line 211: immunohistochemical or immunofluorescent staining? For Fig. 3: how about the staining results of MT-hTERT cells? Fig. 3: any difference of staining results for OCT3/4, SOX2, Nanog, nestin etc. for 3 different strains? Paragraph 4.2: Why HDDPCs should be cultured in gelatin-coated dishes? Purposes? What source and property of the P19 cells can be briefly described? For section 4.10: how about the staining results of DSPP? DSPP expression perhaps is low? For cresyl violet solution staining, is this staining specific?

Author Response

Page 6: Both MT E7 and MT hTERT exhibited similar rates of cell proliferation…… (Fig. 2A). Any analysis of data of 3 strains in Fig. 2A? MT_hTERT seems lower than MT and MT_E7?

Thank you for your careful and comprehensive review. Indeed, we must apologize for the careless mistakes made in the previous version of Fig. 2A. In fact, we counted the cell numbers on days 1, 3, 5, 7, and 9 after seeding rather than counting every day for 5 days as originally indicated. Therefore, we corrected the associated text in the Materials & Methods and legend to Fig. 2A in the revised manuscript (see line 417 and Figure 2A). As explained in P12, we cultivated each group of HDDPCs in five wells of a 24-well plate and counted the cell number. The data are plotted for all three cell lines in Fig. 2A. As indicated by the referee, the proliferation rate of MT_hTERT seems to be lower than that from MT and MT_E7; however, the difference in proliferation rate between MT_hTERT and MT/MT_E7 was not statistically significant (P >0.05). This is now mentioned in the revised text (see lines 183 and 422).

For Fig. 2B: Whether MT_hTERT cells are also shown less ALP activities?

The previous figure was replaced by another as the new Fig. 2B in the revised text, demonstrating positive ALP activity in all three cell lines.

For Fig. 2C: Whether DSPP mRNA expression (a potential expression marker of odontoblasts) is easily identified by RT-PCR? What is the culture condition of cells (medium or mineralized medium? For how many days?)

It is possible to detect DSPP mRNA expression in HDDPCs by RT-PCR, as shown in Fig. 2C. In this case, the HDDPCs used for RT-PCR were cultured in Dulbecco’s modified Eagle’s medium with 20% heat-inactivated fetal bovine serum until reaching 90% confluency in 60-mm gelatin-coated dishes. The cells were harvested via standard trypsinization and subjected to lysis for mRNA preparation. These details have now been added to the revised text (see line 453).  

Page 8: ….exhibited neuronal morphology with elongated axons and dendrites (Fig. 2D) – easily discernable and specific?

  We consider that the cells showing neurite-like extensions that differentiated from HDDPCs after cultivation in mesenchymal stem cell neurogenic differentiation medium (#C-28015; PromoCell, Heidelberg, Germany) are indeed neuronal cells. They are easily identifiable by the presence of cytoplasmic bodies, which are specifically stained with Nissl.

Page 7, line 211: immunohistochemical or immunofluorescent staining? For Fig. 3: how about the staining results of MT-hTERT cells?

  The content mentioned on L211 is about immunofluorescent (not immunohistochemical) staining. This has now been clarified in the revised manuscript (see line 217). The staining results of MT_hTERT were the same as those of the parental cells and MT_E7 cells. Thus, we did not show the results for MT_hTERT cells; however, this point is now mentioned in the revised text (see line 219).

Fig. 3: any difference of staining results for OCT3/4, SOX2, Nanog, nestin etc. for 3 different strains?

There were no appreciable differences in the immunofluorescent staining patterns of anti-OCT3/4, anti-SOX2, anti-NANOG, or anti-NESTIN among the three cell lines.

Paragraph 4.2: Why HDDPCs should be cultured in gelatin-coated dishes? Purposes?

A gelatin-coated plate is superior to a gelatin-free plate with respect to the promotion of cell growth, since gelatin itself can mediate a better cell–substratum interaction. In particular, the primary cultured HDDPCs require such a condition because they do not strongly attach to the dish compared to other continuous cell lines.

What source and property of the P19 cells can be briefly described?

As shown in reference number 46, P19 cells are murine (C3H) teratocarcinoma cells that can differentiate into several types of cells upon differentiation induction, but have no ability to generate functional germ cells.

For section 4.10: how about the staining results of DSPP? DSPP expression perhaps is low?

  All three cell lines (MT, MT_E7, and MT_hTERT cells) were positively stained when immunostaining was performed using anti-DSPP. This point is now mentioned in the revised text (see line 220).  

For cresyl violet solution staining, is this staining specific?

Staining of cells in the presence of cresyl violet is essential for the detection of neuronal cells with Nissl staining.

Reviewer 2 Report

The paper by Inada E. et al. is an elegant demonstration of the non-tumorigenic potential of human deciduous dental pulp stem cells (HDDPSCs) by permanent transfection with a piggy-bac vector system. The main findings of this study are consistent with other related studies in the dental field.

While the experimental procedures and results are solid and innovative, there are several major issues that need to be addressed before the manuscript is ready for publication.

Major issue #1. The identity of the parental HDPPSC lines is uncertain. In the manuscript, the authors refer to another paper (lines 88-89). When checking this other paper (J Invest Clin Dent. 2017;8:e12236) I do not find the acronyms cited but rather the numbers #1-5. It is uncertain to what clone the authors refer to as the parental line MT. Is this the same line they used in the 2017 paper? Why was this parental cell line chosen among the rest?

Major issue #2. Figure 2A. The specified time frame in the graph (0-5 days) is too short to claim there are not differences in proliferation between the parental and transfected cells. The population doubling rate should be estimated over a much longer period (at least several weeks). Additionally, the total number of passages of the MT parental cell should be specified. The authors state in the text that the MT line showed a proliferation arrest at passages 14-16 (lines 89-90). I presume the experiments were performed at lower passages, because the data of Figure 2A still show an important proliferative activity for the parental MT line. Is the transfection with hTERT and E7 capable for rescuing proliferative ability of aged (>P14-16) HDDPSCs.

Major issue #3. Figure 2D. Nissl staining can be hardly considered to be a good assay for neuronal differentiation as this staining labels RNA, which can be found in any other cell type. The authors should switch to immunostaining for other specific neuronal markers to sustain their claims. MAP2, NeuN and/or Douclecortin (DCX) are suggested.

Major issue #4. There is an important concern here regarding ALP activity. This activity is associated with stemness and pluripotency in ESCs and IPSCs, but also with differentiation commitment to osteoblastic cells in bone marrow and dental pulp stem cells. Is there a way to distinguish between both types of ALP activity?

I suggest the authors perform a simple but key control experiment to discard that the detected ALP in your system is not showing the osteoblastic commitment of HDDPSCs, instead of their stemness. Just perform an incubation of your HDDPSC lines with Dexa, B-glycerol-phosphate and ascorbate for 2-3 weeks, and assess ALP activity thereafter. For details on the protocol and reagent concentrations see: Langenbach F, Handschel J (2013) Stem Cell Res Ther 4: 117.

If an increased ALP activity was detected after this osteoblastic differentiation protocol in the control and transfected HDDPSC lines, the authors might have to radically change the interpretation of the corresponding part on the discussion section.

Author Response

Major issue #1. The identity of the parental HDPPSC lines is uncertain. In the manuscript, the authors refer to another paper (lines 88-89). When checking this other paper (J Invest Clin Dent. 2017;8:e12236) I do not find the acronyms cited but rather the numbers #1-5. It is uncertain to what clone the authors refer to as the parental line MT. Is this the same line they used in the 2017 paper? Why was this parental cell line chosen among the rest?

The parental HDPPSC line used in this study is the same as that described in our previous paper (J Invest Clin Dent, 2017; 8: e12236) as line #5. This cell line expresses stem-cell-specific factors, as exemplified by OCT3/4, and could be easily reprogrammed to form iPS cells upon transfection with Yamanaka factors. The immortalization of cell line #5 line allowed us to use this line as a “continuous cell line”, which will be beneficial for further characterization at the molecular level. We have now clarified that the present cell line was that described as cell line #5 in our previous paper in the revised manuscript (see line 89).

Major issue #2. Figure 2A. The specified time frame in the graph (0-5 days) is too short to claim there are not differences in proliferation between the parental and transfected cells. The population doubling rate should be estimated over a much longer period (at least several weeks). Additionally, the total number of passages of the MT parental cell should be specified. The authors state in the text that the MT line showed a proliferation arrest at passages 14-16 (lines 89-90). I presume the experiments were performed at lower passages, because the data of Figure 2A still show an important proliferative activity for the parental MT line. Is the transfection with hTERT and E7 capable for rescuing proliferative ability of aged (>P14-16) HDDPSCs.

First, we must apologize for the careless errors in the original Fig. 2A. In fact, we counted the cell number on days 1, 3, 5, 7, and 9 after seeding rather than counting cells every day for 5 days. Therefore, we corrected the related description in the Materials & Methods and legend to Fig. 2A in the revised manuscript (see line 417 and Figure 2A). The parental HDPPSC line used in this study was passaged for six generations (see line 374). In the proliferation assay, we used a 24-well plate. The resulting immortalized cells have the ability to proliferate continuously, since when these cells are collected 9 days after cell seeding and re-plated onto another new plate, they begin to proliferate actively. This was confirmed with at least 10 replicate experiments. This point is now mentioned in the revised manuscript (see line 419). With respect to the immortalization of aged HDDPCs, it was impossible to obtain immortalized cells, since the cell growth was found to be very poor after transfection.

Major issue #3. Figure 2D. Nissl staining can be hardly considered to be a good assay for neuronal differentiation as this staining labels RNA, which can be found in any other cell type. The authors should switch to immunostaining for other specific neuronal markers to sustain their claims. MAP2, NeuN and/or Douclecortin (DCX) are suggested.

The reagents [mesenchymal stem cell neurogenic differentiation medium (#C-28015; PromoCell, Heidelberg, Germany)] used for the differentiation induction of HDDPCs to neuronal cells used in this study are widely employed for generating neuronal cells from various types of non-neuronal cells. Nissl staining has also been traditionally used to study the morphology and pathology of neural tissues, and is recognized as a valuable tool for the detection of neuronal cells. Based on this background, we considered that the Nissl-positive cells generated after differentiation induction of HDDPCs were neuronal cells. Notably, in our previous experiments in which porcine embryonic fibroblasts were induced to differentiate into neuronal cells by incubating in mesenchymal stem cell neurogenic differentiation medium, no Nissl-positive cells were detected. Thus, the present data in which the Nissl-positive cells were generated after differentiation induction of HDDPCs demonstrated that HDDPCs are able to differentiate into neuronal cells.

Major issue #4. There is an important concern here regarding ALP activity. This activity is associated with stemness and pluripotency in ESCs and IPSCs, but also with differentiation commitment to osteoblastic cells in bone marrow and dental pulp stem cells. Is there a way to distinguish between both types of ALP activity?

I suggest the authors perform a simple but key control experiment to discard that the detected ALP in your system is not showing the osteoblastic commitment of HDDPSCs, instead of their stemness. Just perform an incubation of your HDDPSC lines with Dexa, B-glycerol-phosphate and ascorbate for 2-3 weeks, and assess ALP activity thereafter. For details on the protocol and reagent concentrations see: Langenbach F, Handschel J (2013) Stem Cell Res Ther 4: 117.

If an increased ALP activity was detected after this osteoblastic differentiation protocol in the control and transfected HDDPSC lines, the authors might have to radically change the interpretation of the corresponding part on the discussion section.

We greatly appreciate this relevant question and helpful suggestion of additional experiments. The system that we used to detect ALP activity relies on cytochemical staining using an ALP detection kit, which can detect the activity of every type of ALP. In our previous study, we confirmed that our HDDPCs express a non-tissue-specific form of ALP together with other stem cell-specific factors such as OCT3/4 (Inada et al., J Invest Clin Dent. 2017;8:e12236). These ALP-positive HDDPCs are more amenable to be reprogrammed by transfection with reprogramming factors than the other ALP-negative HDDPCs (Inada et al., J Invest Clin Dent. 2017; 8: e12236). Furthermore, we demonstrated that repeated transfection of the ALP-negative HDDPCs resulted in an increase in the number of ALP-positive cells (Soda et al., Sci Rep. 2019; 9: 1490). This phenomenon was also observed when human fibroblasts were subjected to repeated transfection with the reprogramming factors (Soda et al., Sci Rep. 2019; 9: 1490). Taken together, we consider that ALP expression is well correlated with the immature state of cells and that ALP itself can be a good marker of an undifferentiated state (Soda et al., Sci Rep. 2019; 9: 149, Inada et al., J Invest Clin Dent. 2017;8:e12236). The HDDPCs can also differentiate into ALP-positive osteoblasts when incubated with osteogenic differentiation medium (#KBDSTC103; DS Pharma, Osaka, Japan). Thus, the system the reviewer proposed for distinguishing between the type of ALP activity might not reflect the pluripotency of HDDPCs.   

Round 2

Reviewer 2 Report

Major issue #1. The identity of the parental HDPPSC lines is uncertain. In the manuscript, the authors refer to another paper (lines 88-89). When checking this other paper (J Invest Clin Dent. 2017;8:e12236) I do not find the acronyms cited but rather the numbers #1-5. It is uncertain to what clone the authors refer to as the parental line MT. Is this the same line they used in the 2017 paper? Why was this parental cell line chosen among the rest?

The parental HDPPSC line used in this study is the same as that described in our previous paper (J Invest Clin Dent, 2017; 8: e12236) as line #5. This cell line expresses stem-cell-specific factors, as exemplified by OCT3/4, and could be easily reprogrammed to form iPS cells upon transfection with Yamanaka factors. The immortalization of cell line #5 line allowed us to use this line as a “continuous cell line”, which will be beneficial for further characterization at the molecular level. We have now clarified that the present cell line was that described as cell line #5 in our previous paper in the revised manuscript (see line 89).

Reviewer Response: OK. It is clarified now.

Major issue #2. Figure 2A. The specified time frame in the graph (0-5 days) is too short to claim there are not differences in proliferation between the parental and transfected cells. The population doubling rate should be estimated over a much longer period (at least several weeks). Additionally, the total number of passages of the MT parental cell should be specified. The authors state in the text that the MT line showed a proliferation arrest at passages 14-16 (lines 89-90). I presume the experiments were performed at lower passages, because the data of Figure 2A still show an important proliferative activity for the parental MT line. Is the transfection with hTERT and E7 capable for rescuing proliferative ability of aged (>P14-16) HDDPSCs.

First, we must apologize for the careless errors in the original Fig. 2A. In fact, we counted the cell number on days 1, 3, 5, 7, and 9 after seeding rather than counting cells every day for 5 days. Therefore, we corrected the related description in the Materials & Methods and legend to Fig. 2A in the revised manuscript (see line 417 and Figure 2A). The parental HDPPSC line used in this study was passaged for six generations (see line 374). In the proliferation assay, we used a 24-well plate. The resulting immortalized cells have the ability to proliferate continuously, since when these cells are collected 9 days after cell seeding and re-plated onto another new plate, they begin to proliferate actively. This was confirmed with at least 10 replicate experiments. This point is now mentioned in the revised manuscript (see line 419). With respect to the immortalization of aged HDDPCs, it was impossible to obtain immortalized cells, since the cell growth was found to be very poor after transfection.

Reviewer Response: OK. I think the manuscript improves with these clarifications.

Major issue #3. Figure 2D. Nissl staining can be hardly considered to be a good assay for neuronal differentiation as this staining labels RNA, which can be found in any other cell type. The authors should switch to immunostaining for other specific neuronal markers to sustain their claims. MAP2, NeuN and/or Douclecortin (DCX) are suggested.

The reagents [mesenchymal stem cell neurogenic differentiation medium (#C-28015; PromoCell, Heidelberg, Germany)] used for the differentiation induction of HDDPCs to neuronal cells used in this study are widely employed for generating neuronal cells from various types of non-neuronal cells. Nissl staining has also been traditionally used to study the morphology and pathology of neural tissues, and is recognized as a valuable tool for the detection of neuronal cells. Based on this background, we considered that the Nissl-positive cells generated after differentiation induction of HDDPCs were neuronal cells. Notably, in our previous experiments in which porcine embryonic fibroblasts were induced to differentiate into neuronal cells by incubating in mesenchymal stem cell neurogenic differentiation medium, no Nissl-positive cells were detected. Thus, the present data in which the Nissl-positive cells were generated after differentiation induction of HDDPCs demonstrated that HDDPCs are able to differentiate into neuronal cells.

Reviewer R: I am afraid I have to disagree here. Although sometimes accepted (and so published) as a neuron-specific staining, Nissl labeling is not specific to neuronal cells. Many other cell types stain with Nissl, including glia and endothelial cells, among many others. Thus it is very uncertain that those Nissl-positive cells are indeed neuron-related cells, or something else. I think this is a very widespread problem with scientific literature about neuronal differentiation from non-neural stem cells, and not something specific to this particular paper. We authors sometimes adopt too lax criteria to define a neuronal cell. This in my opinion contributes in no-small amount to arise skepticism among neuroscientists who believe these studies are inherently flawed, and with good reason, as no detailed characterization of the obtained cell type is performed. Even immunostaining for some supposedly specific neuronal proteins is inherently biased since some of them (e.g. Nestin, Tuj-1, Neurofilament-H) are constitutively expressed by hDPSCs. I insist that the authors perform and add some specific immunostaining for MAP2, DCX and/or NeuN, and possibly include s100b as well, to confirm the neuronal/glial phenotype of the obtained cells. In addition, unless some functional assays (e.g. electrophysiology) are performed, authors should not describe the obtained cells as neuronal cells, but “neuronal-like” cells.

Major issue #4. There is an important concern here regarding ALP activity. This activity is associated with stemness and pluripotency in ESCs and IPSCs, but also with differentiation commitment to osteoblastic cells in bone marrow and dental pulp stem cells. Is there a way to distinguish between both types of ALP activity?

I suggest the authors perform a simple but key control experiment to discard that the detected ALP in your system is not showing the osteoblastic commitment of HDDPSCs, instead of their stemness. Just perform an incubation of your HDDPSC lines with Dexa, B-glycerol-phosphate and ascorbate for 2-3 weeks, and assess ALP activity thereafter. For details on the protocol and reagent concentrations see: Langenbach F, Handschel J (2013) Stem Cell Res Ther 4: 117.

If an increased ALP activity was detected after this osteoblastic differentiation protocol in the control and transfected HDDPSC lines, the authors might have to radically change the interpretation of the corresponding part on the discussion section.

We greatly appreciate this relevant question and helpful suggestion of additional experiments. The system that we used to detect ALP activity relies on cytochemical staining using an ALP detection kit, which can detect the activity of every type of ALP. In our previous study, we confirmed that our HDDPCs express a non-tissue-specific form of ALP together with other stem cell-specific factors such as OCT3/4 (Inada et al., J Invest Clin Dent. 2017;8:e12236). These ALP-positive HDDPCs are more amenable to be reprogrammed by transfection with reprogramming factors than the other ALP-negative HDDPCs (Inada et al., J Invest Clin Dent. 2017; 8: e12236). Furthermore, we demonstrated that repeated transfection of the ALP-negative HDDPCs resulted in an increase in the number of ALP-positive cells (Soda et al., Sci Rep. 2019; 9: 1490). This phenomenon was also observed when human fibroblasts were subjected to repeated transfection with the reprogramming factors (Soda et al., Sci Rep. 2019; 9: 1490). Taken together, we consider that ALP expression is well correlated with the immature state of cells and that ALP itself can be a good marker of an undifferentiated state (Soda et al., Sci Rep. 2019; 9: 149, Inada et al., J Invest Clin Dent. 2017;8:e12236). The HDDPCs can also differentiate into ALP-positive osteoblasts when incubated with osteogenic differentiation medium (#KBDSTC103; DS Pharma, Osaka, Japan). Thus, the system the reviewer proposed for distinguishing between the type of ALP activity might not reflect the pluripotency of HDDPCs.

Reviewer Response: It is curious that the authors appreciate my helpful suggestion of just one additional experiment but nevertheless decline to perform it. I am afraid I cannot let it pass because this is a very relevant question concerning the phenotype of these cells. Most authors in the dental stem cell field associate positivity to ALP with osteoblastic pre-differentiation, and here I see authors to claim just the contrary, that ALP-positive cells are the ones looking more stem-like. In my opinion, it is imperative that we solve this discrepancy, by exposing your HDPPC lines to osteoblastic differentiation reagents and see whether ALP activity gets affected or not. It is important to point out that these dental pulp cells differentiate spontaneously to osteoblastic cells in the presence of FBS in similar conditions to the ones of this paper (Pisciotta et al. PlosOne 2012; 7(11):e50542; Yu et al. BMC Cell Biology 2010, 11:32). Thus, I think it is very necessary to address this issue. The idea is not to reflect the pluripotency of HDDPCs, is just to check whether the ALP staining could also be indicative of differentiation to osteoblastic/odontoblastic cells in your experimental conditions. The positivity to DSPP, as you describe in new line 220, seems also to add support to this alternative view of your results, which demands a note of caution. Thus, I would demand the authors to please also address the ALP activity of your cell lines when incubated under the Osteogenic differentiation medium, as described in line 513 (#KBDSTC103; DS Phama, Osaka, Japan) for 14 days. In addition, photometric quantification of ALP absorbance should be made in the different cell lines under control and osteogenic stimulation conditions, and include these data together with ALP images.

Thus, this reviewer would be ready to endorse the manuscript provided the necessary control experiments and amendments are performed, regarding the pending major issues #3 and #4.